**METHOD**

# Knockout of circRNAs by base editing back-splice sites of circularized exons

Xiang Gao[1,2†], Xu-Kai Ma[3†], Xiang Li[1,4], Guo-Wei Li[3], Chu-Xiao Liu[1], Jun Zhang[1], Ying Wang[1], Jia Wei[3], Jia Chen[2,5], Ling-Ling Chen[1,4] and Li Yang[3*]

* Correspondence: liyang@picb.ac.cn
†Xiang Gao and Xu-Kai Ma contributed equally to this work.
3CAS Key Laboratory of Computational Biology, Shanghai Institute of Nutrition and Health , University of Chinese Academy of Sciences, Chinese Academy of Sciences, 320 Yueyang Road, Shanghai 200031, China
Full list of author information is available at the end of the article

## Abstract

Many circular RNAs (circRNAs) are produced from back-splicing of exons of precursor mRNAs and are generally co-expressed with cognate linear RNAs. Methods for circRNA-specific knockout are lacking, largely due to sequence overlaps between forms. Here, we use base editors (BEs) for circRNA depletion. By targeting splice sites involved in both back-splicing and canonical splicing, BEs can repress circular and linear RNAs. Targeting sites predominantly for circRNA biogenesis, BEs could efficiently repress the production of circular but not linear RNAs. As hundreds of exons are predominantly back-spliced to produce circRNAs, this provides an efficient method to deplete circRNAs for functional study.

**Keywords:** Circular RNA, Base editor, Predominantly back-spliced exon, Splicing, Back-splicing, Knockout

## Introduction

Different from canonical splicing that links an upstream 5′ splice site (ss) with a downstream 3′ ss, back-splicing joins a downstream 5′ back-splice site (bss) with an upstream 3′ bss to produce covalently closed circular RNAs (circRNAs) [1–7]. Despite of being unfavorably processed, back-splicing is catalyzed by the same spliceosomal machinery as canonical splicing [8–10], suggesting their direct competition [11]. In addition, back-splicing is also tightly regulated by *cis*-elements and *trans*-factors [10, 12–16], leading to a spatiotemporal expression of circRNAs across a wide spectrum of examined cell lines, tissues, and species [17–25]. Increasing lines of evidence have now shown that dysregulation of circRNA expression is associated with human diseases, such as cancer [26–29], systemic lupus erythematosus [30], and neuronal degeneration [31, 32], suggesting their potential roles in both physiological and pathological conditions [1, 2, 5]. Mechanically, most circRNAs are localized in cytosol and some were found to act as decoys for miRNAs or proteins [12, 15, 19, 22, 30, 32, 33].

Nevertheless, biological significance of most circRNAs remains largely unexplored, partially due to limited methods for their functional studies, such as circRNA knockout (KO) at the DNA level. For example, the CRISPR/Cas9 genome editing removed the

entire back-spliced exon to produce *Cdr1as/ciRS-7* KO mouse, which showed defects in excitatory synaptic transmission [32]. However, since the CRISPR/Cas9 KO method results in a large fragment deletion and that sequences of circular and their cognate linear RNAs are generally overlapped, CRISPR/Cas9-mediated circRNA KO could inevitably impair linear parental transcripts and is not appropriate for conducting large-scale screening. It is applicable to *Cdr1as/ciRS-7*, mainly because of the predominant expression of *Cdr1as/ciRS-7* with little if any expression of its linear cognate RNAs [19, 22, 32, 34]. In addition, as mammalian circRNA biogenesis is generally facilitated by intronic complementary sequences (ICSs) flanking back-spliced exon(s), an alternative circRNA KO strategy is via the disruption of pairing of ICSs. Previously, the human *circGCN1L1* was knocked out without affecting the linear RNA expression in PA1 cells by deleting one side of ICSs flanking back-spliced exons [10]. However, given that circRNA biogenesis regulated by ICSs is complicated and multiple ICSs are often involved in circRNA biogenesis [16, 35, 36], this indirect KO strategy is inadequate at most circRNA-producing loci with several pairs of ICSs [5, 10]. A simple and efficient method for circRNA KO has long been desired.

Recently, a rich arsenal of base editors (BEs) that combine different types of nucleobase deaminases with distinct CRISPR/Cas proteins have been developed to achieve targeted C-to-T (CBE) or A-to-G (ABE) changes at single-nucleotide resolution [37–40]. Given their efficiency, specificity, and safety, BEs are believed to have broad applications in both basic research and therapeutics [39–41]. Specifically, mutating nucleotide sequences at canonical splice sites by BEs has been used for altering splicing patterns [42, 43]. Inspired by these findings, we sought to apply BEs to target back-splice sites for endogenous circRNA KO at the genomic level. Here, we showed that BEs repressed both circular and linear RNAs expression at the same gene loci when targeting splice sites simultaneously involved in back-splicing and canonical splicing, confirming the requirement of the same splice site signals for back-splicing and canonical splicing in vivo. Differently, by targeting sites predominantly for back-splicing, a set of circRNAs, including *CDR1as/ciRS-7*, were specifically abolished without obvious effects on the expression of their cognate linear RNAs. We further applied BEs for a small-scale loss-of-function (LOF) screening of circRNAs and found a circRNA with previously unannotated exon in the *ZNF292* gene locus that represses cell proliferation. Collectively, our results confirm the requirement of canonical splice signals for both canonical splice and back-splice at the genomic DNA level and demonstrate an efficient and specific method for endogenous circRNA KO with BEs.

## Results

### Design of applying BEs to knock out circular RNA expression

Genome-wide analysis revealed that nearly identical consensus sequences existed between 5′ ss and 5′ bss or between 3′ ss and 3′ bss (Fig. 1A), consistent with previous finding that the same spliceosomal machinery is required for back-splicing [8–10]. Specifically, genomic sequences of AG/gt are enriched at exon/intron junctions of both 5′ ss and 5′ bss, and ag/GT are enriched at intron/exon junctions of both 3′ ss and 3′ bss. Given that genomic splice site mutagenesis by BEs could change splicing patterns [42, 43], we hypothesized that genomic sequences at back-splice sites could be targeted

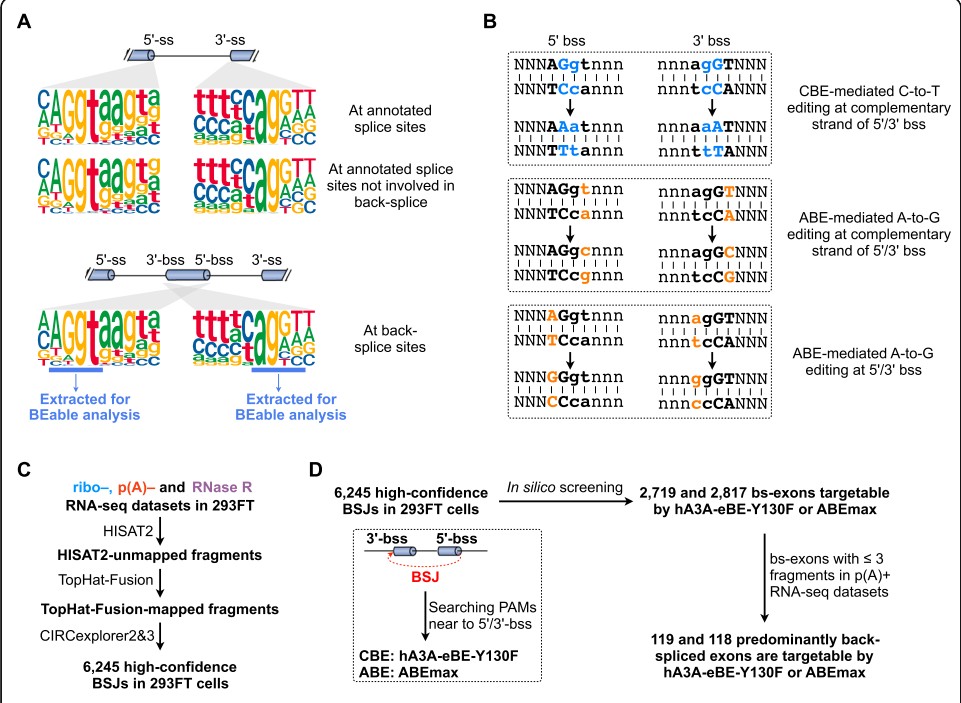

**Fig. 1** Consensus sequence analysis of (back-)splice sites and prediction of their availability to base editing. **A** Genome-wide analyses of consensus sequences at all 5′/3′ splice sites (5′/3′ ss), 5′/3′ ss without back-splice (top), or 5′/3′ back-splice sites (5′/3′ bss) (bottom) of annotated exons. Ten bases around 5′ bss/5′ ss (three upstream exonic bases and seven downstream intronic bases) and ten bases around 3′ bss/3′ ss (seven upstream intronic bases and three downstream exonic bases) were fetched for consensus sequence analysis. Intronic sequences were represented by a, t, c, and g, and exonic sequences were represented by A, T, C, and G. **B** Diagram of directing base editor (BE) to introduce base mutation at 5′/3′ bss. CBE could lead to C-to-T base editing at complementary strands of 5′/3′ bss. ABEs could introduce A-to-G base editing at 5′/3′ bss or at their complementary strands. **C** Prediction of circRNAs from ribo−, p(A)−, and RNaseR RNA-seq datasets from 293FT. **D** In silico screening of circRNAs with predominantly back-spliced exons could be targeted by hA3A-eBE-Y130F or ABEmax at back-splice sites

by BEs to potentially disrupt back-splice (Fig. 1B). In principle, both CBEs and ABEs are applicable for nucleotide changes at exon/intron junctions of back-splice sites. On the one hand, cytosine(s), which is/are base-paired at the complementary strand to guanine(s) at exon/intron junctions of back-splice sites, could be changed to thymine(s) by CBEs (top, Fig. 1B and Additional file 1: Fig. S1). On the other hand, ABEs could introduce A-to-G base editing at 5′/3′ bss or at their complementary strands (bottom, Fig. 1B and Additional file 1: Fig. S2). By searching for nearby PAM sequences to fit the targeted bases into editing windows of selected BEs (Additional file 1: Fig. S1 and S2) [44], both cases could introduce base substitution(s) at back-splice sites, which would deplete circRNA production in principle.

We set up to computationally predict back-splice sites that could be targeted by BEs [44]. By identifying RNA-seq fragments mapped to back-splice junction (BSJ) sites [16, 21, 36], back-splice sites spanning BSJs and their corresponding circRNAs were profiled from transcriptomic datasets of human 293FT cells (Fig. 1C). Genomic sequences of AG/gt at exon/intron junctions of 5′ bss and ag/GT at intron/exon junctions of 3′ bss were then extracted to examine the accessibility by three CBEs (including hA3A-eBE-Y130F [45], BE4max [46] and eBE-S3 [47]) and one ABE (ABEmax [46]). Of note, since the same SpCas9 nickase (nCas9) was used for the construction of hA3A-eBE-Y130F,

BE4max, and eBE-S3, their targeted 5′ bss/3′ bss were almost overlapped with slight difference due to their varied editing windows. By identifying nearby PAM motifs that could guide selected BEs to introduce base changes at exon/intron (or intron/exon) junctions, thousands of back-spliced exons were predicted to be target candidates of BEs (Fig. 1D and Additional file 2: Table S1) which implies the potential broad application of this method for circRNA KO.

### Mutating splice site sequences of exons involved in both back-splice and canonical splice by BEs abolishes both circular and linear RNA expression

Next, we applied BEs to test whether introducing base changes at exon/intron (or intron/exon) junctions of back-spliced exons could knock out circRNAs at the genomic level (Fig. 2A). We started with two BE-targeted 5′ (b)ss of exons at *SPECC1* and *FNTA* gene loci, and additional two BE-targeted 3′ (b)ss of exons at *FOXP1* and *ZCCHC2* gene loci. Specific sgRNAs were designed to fit the targeted cytosines at the complementary strand to exon/intron junction of back-splice sites into the editing windows of BEs [44]. Of note, these exons were involved in both back-splice for circRNAs and canonical splice for linear RNAs in 293FT cells (top, Fig. 2B−E). After transfecting 293FT cells with vectors for a specific BE and a corresponding sgRNA, genomic DNAs and total RNAs were individually extracted to evaluate base editing efficiency at BE-targeted sites and its corresponding effect on circRNA and linear RNA biogenesis.

Genomic DNA amplification and subsequent Sanger sequencing showed that hA3A-eBE-Y130F achieved ∼ 40–70% G-to-A, complementary C-to-T, changes at all four targeted 5′ or 3′ (b)ss in the condition of transient transfection (middle, Fig. 2B−E).

Correspondingly, efficiencies of back-splice of all four targeted exons were reduced to comparable levels to G-to-A base editing efficiencies (bottom, Fig. 2B−E). As expected, canonical splice levels of these targeted exons were also repressed to comparable levels of back-splice alteration (bottom, Fig. 2B-E). In addition, when amplifying fragments of linear transcripts that were far away from edited 5′ or 3′ (b)ss, it also suggested an observed, but less, reduction of cognate linear RNA expression (labeled with "_down", Fig. 2B−E), possibly due to the decay of mis-spliced linear RNAs [48]. Of note, different editing efficiencies of nearby guanines (which are base-paired at the complementary strand to cytosines targeted by BEs) at exon/intron junctions of back-splice sites were observed, consistent to previous results that different editing efficiencies of nearby cytosines within a given editing window [45]. This can be due to different accessibilities by the deaminase moiety of BEs for deamination reaction, different contexts of targeted cytosines, and/or methylation levels (high or low) of targeted cytosines [45].

A variety of BEs has been developed to catalyze base changes with different efficiencies and specificities [40, 41, 44]. Then, we tested other two CBEs, BE4max [46] and eBE-S3 [47], on their base editing effects at the targeted 5′ (b)ss of exon at *SPECC1* gene locus and the targeted 3′ (b)ss of exon at *FOXP1* gene locus (Additional file 1: Fig S1). Similar to results obtained by using hA3A-eBE-Y130F, BE4max and eBE-S3 led to 40–60% base change at intended 5′ (b)ss of *circSPECC1* or 3′ (b)ss of *circFOXP1* (middle, Additional file 1: Fig. S1C and S1D), together with a comparable reduction of back-splice and cognate linear RNA splice (bottom, Additional file 1: Fig. S1C and S1D). In addition, we also used ABEmax [46] to directly introduce A-to-G editing,

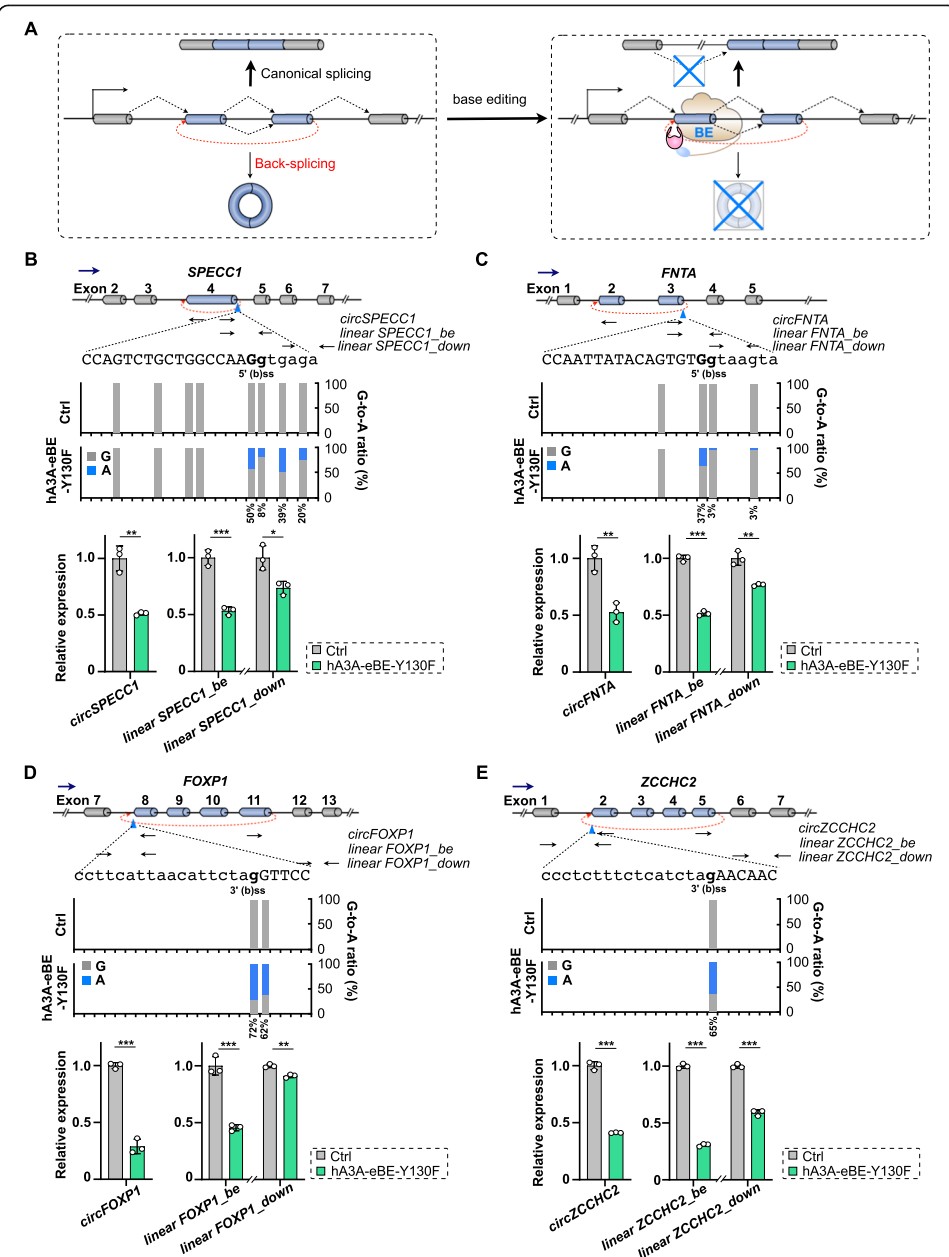

**Fig. 2** Base editing at back-splice sites generally leads to alternation of back-splice and canonical splice. **A** Schematic of base editing and its outcomes at splice sites involved in both back-splice and canonical splice. **B** Base changes at 5′ (b)ss of *cirSPECC1* by hA3A-eBE-Y130F repressed both back-splice for circRNAs and canonical splice for linear RNA expression. Top, schematic of partial *SPECC1* gene organization. Back-spliced exon 4 was highlighted by blue bar. Context sequences of targeted 5′ (b)ss were shown by a, t, c, and g for intron or by A, T, C, and G for exon; Middle, G-to-A base change ratio at targeted 5′ (b)ss of back-spliced exon 5 in the *SPECC1* gene locus; Bottom, evaluation of back-splice and splice changes by RT-qPCR using primers labeled on the top. **C** Base changes at 5′ (b)ss of *circFNTA* by hA3A-eBE-Y130F repressed both back-splice for circRNAs and canonical splice for linear RNA expression. Refer to **B** for details. **D** Base changes at 3′ (b)ss of *circFOXP1* by hA3A-eBE-Y130F repressed both back-splice for circRNAs and canonical splice for linear RNA expression. Top, schematic of partial *FOXP1* gene organization. Back-spliced exons 8-11 were highlighted by blue bars. Context sequences of targeted 3′ (b)ss were shown by a, t, c, and g for intron or by A, T, C, and G for exon; Middle, G-to-A base change ratio at targeted 3′ (b)ss of back-spliced exon 8 in the *FOXP1* gene locus; Bottom, evaluation of back-splice and splice changes by RT-qPCR using primers labeled on the top. **E** Base changes at 3′ (b)ss of *circZCCHC2* by hA3A-eBE-Y130F repressed both back-splice for circRNAs and canonical splice for linear RNA expression. Refer to **D** for details. **B–E** Error bar represents SD from three independent replicates. ∗, $P <$ 0.05; ∗∗, $P < 0.01$; ∗∗∗, $P < 0.001$, Student's *t* test

complementary to T-to-C changes at the same targeted 5′ (b)ss of *circSPECC1* or 3′ (b)ss of *circFOXP1*. As a result, ABEmax introduced ∼ 50% T-to-C/A-to-G mutation at 5′ (b)ss of *circSPECC1* or 3′ (b)ss of *circFOXP1*, and a similar reduction of back-splice and cognate linear RNA splice was observed as those by CBEs (Additional file 1: Fig. S2).

Together, these findings confirmed to use BE for circRNA KO at the genomic DNA level. However, due to the requirement of the same splice signals for both back-splice and canonical splice in mammalian cells (Fig. 2A), simultaneous repression of circular and their corresponding linear cognate RNAs were observed. This further indicated a direct competition between back-splice for circRNAs and canonical splice for linear RNAs between overlapped 5′ ss and 5′ bss or 3′ ss and 3′ bss [11] at real genomic sites.

### Specific knockout of predominantly expressed *CDR1as/ciRS-7* by BE at its gene locus

Although targeting exons involved in both back-splice and canonical splice by BEs could repress both circular and linear RNA expression, targeting exons that are predominantly back-spliced for circRNA formation by BEs could theoretically achieve specific KO effect on circRNAs. To test this speculation, we first set to manipulate 5′ bss of *CDR1as/ciRS-7* for its potential KO by BEs. *CDR1as/ciRS-7* is predominantly expressed at its gene locus (Fig. 3A) [19, 22, 49], evidenced by the fact that multiple cognate linear transcripts originated from both strands were much less expressed than *CDR1as/ciRS-7* [19, 22, 34, 49]. Previously, loss-of-function study of mouse *Cdr1as/ciRS-7* was achieved by using CRISPR/Cas9 genome-editing system to remove the entire back-spliced *Cdr1as/ciRS-7* exon [32]. Here, we tempted to apply BEs to change a few genomic sequences at the 5′ bss of *CDR1as/ciRS-7*, which is distinct from removing the whole circularized exon for *Cdr1as/ciRS-7* KO [32].

After treating with hA3A-eBE-Y130F [44, 45] and corresponding sgRNAs in 293FT cells, two guanines, which are base-paired to two cytosines at the complementary strand of the 5′ bss of *CDR1as/ciRS-7*, were successfully changed to adenines (Fig. 3B). Correspondingly, back-splice of *CDR1as/ciRS-7* was dramatically repressed in the condition of transfected cell mixture (Fig. 3C), suggesting a successful KO effect of endogenous *CDR1as/ciRS-7* by base editing its back-splice site. To further examine this effect, we selected monoclones from BE-treated mixture cells, together with negative control monoclones for parallel comparison. Targeted genomic DNA amplification and Sanger sequencing showed that sequences at the 5′ bss of *CDR1as/ciRS-7* in four out of twenty monoclones were successfully edited as expected (Fig. 3D). The 100% G-to-A (complementary C-to-T) change at the exon end of the 5′ bss of *CDR1as/ciRS-7* was observed at all of three alleles in all four monoclones 1#–4#, suggesting a complete base change at the 5′ bss of *CDR1as/ciRS-7*. Meanwhile, ∼ 67% G-to-A (complementary C-to-T) change at the genomic intron end of the 5′ bss of *CDR1as/ciRS-7* existed at two out of three alleles in 293FT monoclones 2# and 4# (Fig. 3D). In all four monoclones with intended base editing at the 5′ bss of *CDR1as/ciRS-7*, *CDR1as/ciRS-7* expression was barely detected by both Northern blotting and RT-qPCR (Fig. 3E and F).

Similar to previous reports [18, 19, 22, 32], *CDR1as/ciRS-7* was also expressed significantly higher (∼ 100 fold) than its cognate linear RNAs in 293FT cells, shown by RT-qPCR (gray bars, Fig. 3F). Notably, both expression of the *LINC00632* precursor (*pre*, Fig. 3F) and linear cognate RNA, *LINC00632_s1* [34] were not repressed upon

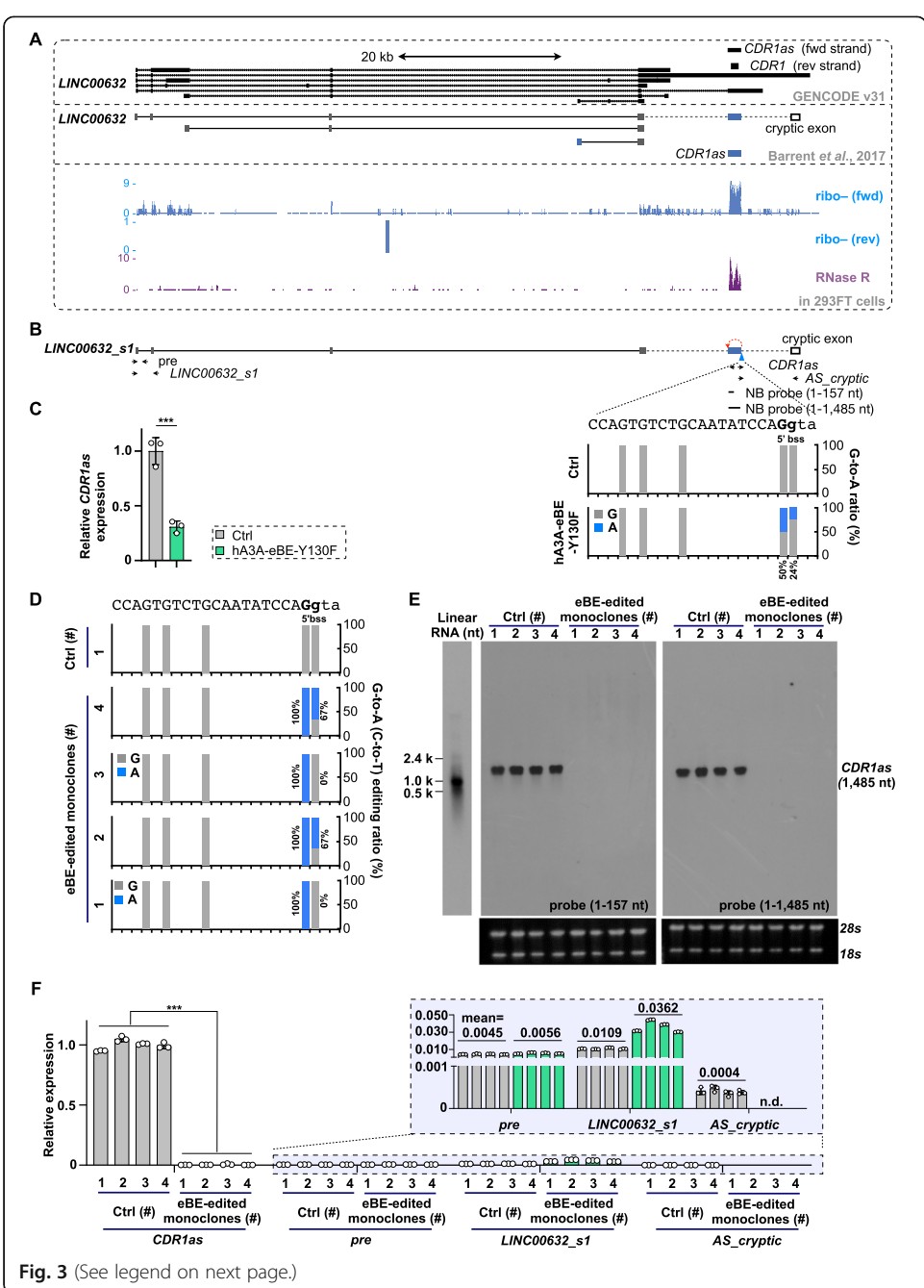

**Fig. 3** (See legend on next page.)

(See figure on previous page.)

**Fig. 3** Achievement of *CDR1as/ciRS-7* knockout by base editing at its 5′ back-splice site. **A** Schematic of the *CDR1as/ciRS-7* gene organization and mapped RNA-seq signals in 293FT cells. Top, multiple transcripts were predicted/reported in the *CDR1as/ciRS-7* gene locus, including a long noncoding RNA containing *CDR1as/ciRS-7*-residing exon (blue bar) and a cryptic exon (blank bar). Bottom, the circular molecule, *CDR1as/ciRS-7*, was confirmed as the major transcript produced from its gene locus, enriched after RNase R treatment. **B** Design of base changes at 5′ (b)ss of *CDR1as/ciRS-7* by hA3A-eBE-Y130F. Top, schematic of partial *CDR1as/ciRS-7* gene organization. The back-spliced *CDR1as/ciRS-7* (blue bar) and a cryptic exon (blank bar) were reported to be also spliced in a long noncoding RNA. Middle, primers for RT-qPCR and probes for northern blotting. Bottom, context sequences of targeted 5′ (b)ss were shown by a, t, c, and g for intron or by A, T, C, and G for exon; G-to-A base change ratio at targeted 5′ (b)ss of back-spliced *CDR1as/ciRS-7* exon was examined in transfected 293FT cell mixture. **C** Repression of *CDR1as/ciRS-7* back-splice by base changes at its 5′ bss. RT-qPCR was performed with primers labeled in **B**. **D** Selection of monoclones with corresponding base editing changes at the 5′ bss of *CDR1as/ciRS-7*. Four monoclones were identified with almost 100% G-to-A base change at the exon boundary of the *CDR1as/ciRS-7* 5′ (b)ss, and among them, monoclones #2 and #4 have an additional G-to-A change (~ 67%) at the intron boundary of the *CDR1as/ciRS-7* 5′ bss. Four monoclones with unchanged bases at the 5′ bss of *CDR1as/ciRS-7* were used as controls (#1 is showed in this panel). **E** Expression of *CDR1as/ciRS-7* was undetected in the four selected monoclones with base editing changes at the 5′ bss of *CDR1as/ciRS-7*, revealed by northern blotting with two probes (1–157 nt and 1–1485 nt). Total RNAs were denatured and then resolved on 1.5% native agarose gel. **F** Back-splice of *CDR1as/ciRS-7* was barely detected in the four selected monoclones with base editing changes at the 5′ bss of *CDR1as/ciRS-7*, revealed by RT-qPCR. Canonical splice along its cognate linear RNA was further compared by parallel RT-qPCR. n.d. indicates non-detected. **C, F** Error bar represents SD from three independent replicates. ∗∗∗, $P < 0.001$, Student's *t* test

*CDR1as/ciRS7* KO in 293FT cells. In addition, the splice of the *CDR1as/ciRS-7* exon and a downstream cryptic exon that produce the alternative splice (AS) cryptic *LINC00632* RNA (*AS_cryptic*, Fig. 3F) [34] was almost completely inhibited by disrupting 5′ bss of *CDR1as/ciRS-7* because of the overlapping 5′ bss and the 5′ ss. As the splice between the *CDR1as/ciRS-7* exon and the downstream cryptic exon was much less (< 1,000 fold) than the back-splice of *CDR1as/ciRS-7* exon itself in examined 293FT cells (gray bars, Fig. 3F), the occurrence of disrupted splice event and possible affected expression of linear cognate RNAs might have limited effect on future functional evaluation of *CDR1as/ciRS-7* KO. Together, these results thus suggested a convenient method by using base editors to achieve circRNA KO without deleting the full circularized fragments.

## Mutating splice sites of predominantly back-spliced exons with BE depletes corresponding circRNA, but not cognate linear RNA, expression

To further identify circRNA-specific exons for BE-mediated KO as for the case of *CDR1as/ciRS-7* (Fig. 3), we next compared polyadenylated linear RNA transcriptomic datasets with non-polyadenylated ones and identified exons that were predominantly back-spliced for circRNA biogenesis, but rarely spliced for cognate linear RNAs (Fig. 1D) [36]. In 293FT cells, ~ 5% of BE-targetable circularized exons were predominantly back-spliced for circRNAs (Fig. 1D and Additional file 2: Table S1). We envisioned that targeting back-splice sites of these predominantly back-spliced exons could deplete corresponding circRNA expression with little effect on their cognate linear RNA biogenesis (Fig. 4A), as for *CDR1as/ciRS-7* KO shown in Fig. 3.

We then set to apply BEs to introduce base changes at back-splice sites of circRNAs with predominantly back-spliced novel exon(s). Two circRNAs at *RALY* and *CAMK1D* loci, each containing a previously unannotated back-spliced exon, were chosen from

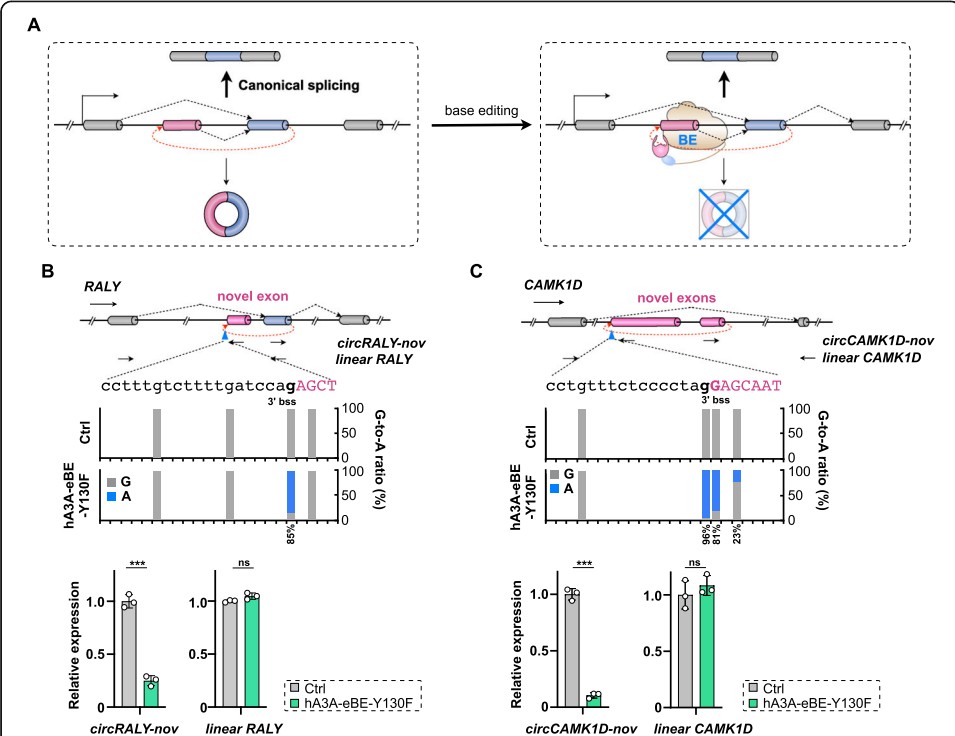

**Fig. 4** Specific circRNA KO with base changes at back-splice sites of predominantly back-spliced exons by BE. **A** Schematic of circRNA knockout by BE-mediated back-splice sites mutation. BEs could introduce base changes at back-splice sites of predominantly back-spliced novel exons (pink bar), resulting in corresponding circRNA knockout with little effect on cognate linear RNA expression. **B** Base changes at 3′ bss of the novel exon in the *RALY* gene locus by hA3A-eBE-Y130F repressed the back-splice for *circRALY-nov*. Top, Schematic of partial *RALY* gene organization. Back-spliced novel exon was highlighted by pink bar. Context sequences of targeted 3′ (b)ss were shown by a, t, c, and g for intron or by A, T, C, and G for exon; Middle, G-to-A base change ratio at targeted 3′ (b)ss of predominantly back-spliced novel exon in the *RALY* gene locus; Bottom, evaluation of back-splice and splice changes by RT-qPCR using primers labeled on the top. Since the identified novel exon was only back-spliced in *circRALY-nov*, base changes at its 3′ bss only affect back-splice of *circRALY-nov*, but not back-splice for canonical splice for linear *RALY* RNA(s) with annotated exons. Error bar represents SD from three independent replicates. *ns*, not significant; ∗∗∗, *P* < 0.001; Student's *t* test. **C** Base changes at 3′ bss of the novel exon in the *CAMK1D* gene locus by hA3A-eBE-Y130F repressed the back-splice for *circCAMK1D-nov*. Refer to **B** for details

293FT transcriptomes for subsequent tests. It should be noted that these two circRNAs with previously unannotated back-spliced exons could be also found in other published datasets [36, 50] (data not shown), referred to as *circRALY-nov* and *circCAMK1D-nov*, respectively.

At the *RALY* locus, the novel exon (159 bp in length) is located between exons 1 and 2, but reversely back-spliced with exon 2 to form *circRALY-nov* (Fig. 4B and Additional file 1: Fig. S3A). At the *CAMK1D* locus, two novel exons (870 bp and 268 bp in length) are located between exons 1 and 2, and back-spliced to form *circCAMK1D-nov* (Fig. 4C and Additional file 1: Fig. S3B). Using divergent primers spanning their BSJs, back-splice of exon 2 and novel exon at the *RALY* gene locus, as well as that of two novel exons at the *CAMK1D* locus, could be successfully identified in both RNase R un-treated and treated RNA samples from 293FT cells (top, Additional file 1: Fig. S3C and S3D). Sanger sequencing of amplified cDNA fragments confirmed these back-splice events for *circRALY-nov* and *circCAMK1D-nov* (middle, Additional file 1: Fig. S3C and S3D). As expected, linear RNA splicing between exon 1 and exon 2 in the *RALY* or

*CAMK1D* gene locus was only detected in the RNase R untreated, but not RNase R treated, RNA samples from 293FT cells with convergent primers (Additional file 1: Fig. S3C and S3D). In both cases, the novel exons were rarely spliced into linear RNAs, evaluated by both the lengths of PCR products and the Sanger sequencing (Additional file 1: Fig. S3C and S3D), which was consistent with the results from RNA-seq datasets (Additional file 1: Fig. S3A and S3B).

As barely spliced in cognate linear RNAs, these back-spliced novel exons were ideal targets of BEs for circRNA-specific KO (Fig. 4A). With sgRNAs targeting the 3′ bss of previously unannotated back-spliced exon in *circRALY-nov* or *circCAMK1D-nov*, more than 80% G-to-A changes were achieved using hA3A-eBE-Y130F for targeted base editing in *RALY* and *CAMK1D* loci (top, Fig. 4B and C), respectively. Correspondingly, back-splice events of *circRALY-nov* or *circCAMK1D-nov* were decreased ~ 70% or 90% (bottom, Fig. 4B and C). Since these novel exons were predominantly back-spliced into circRNAs, but barely spliced into linear RNAs, the splice events (and hence expression) of linear *RALY* and *CAMK1D* RNA transcripts were barely affected (bottom, Fig. 4B and C). Together, these findings suggested a practical and feasible way of applying BEs with sgRNAs targeting back-splice sites of predominantly back-spliced exon(s) for LOF studies of circRNAs.

### Applying BEs for functional circRNA screening

Next, we explored the feasibility of using the BE system for a small-scale LOF screening of circRNAs that contain previously unannotated, back-spliced exons. Among 119 predominantly back-spliced exons that could be targeted by hA3A-eBE-Y130F in 293FT cells (Fig. 1D), 59 of them were previously unannotated in GENCODE annotation (Fig. 5A). In addition, thirteen out of 59 circRNAs were successfully detected in at least two of three (ribo−, polyA−, or RNaseR-treated RNA-seq) datasets from 293FT cells (Fig. 5A), and then subject for function screening. With transfection of vectors for hA3A-eBE-Y130F and designed sgRNA that targets the novel back-splice site of individual circRNAs, effective base mutation (> ~ 50%) at splice sites were obtained at ten out of thirteen cases (Additional file 1: Fig. S4A), and expression of these ten circRNAs was correspondingly suppressed (Additional file 1: Fig. S4A and S4B). To identify circRNAs that may affect cell proliferation, cell proliferation assays were carried out with BE-treated 293FT cells (Fig. 5B). Compared to control treatment, the depletion of *circZNF292-nov* showed an increased effect on cell growth (Fig. 5B), suggesting a repression role of *circZNF292-nov* on cell growth.

Different to a previously reported circRNA, *circZNF292* that consists of three annotated exons (exons 2, 3, and 4) [51], from the *ZNF292* gene locus, *circZNF292-nov* contains three same annotated exons (exons 2, 3, and 4) and one previously unannotated exon between annotated exons 1 and 2 (top, Fig. 5C and Additional file 1: S5A). In addition, the back-splicing of *circZNF292-nov* is processed between the annotated exon 4 and the previously unannotated exon (top, Fig. 5C and Additional file 1: S5A), validated by RT-PCR followed by Sanger sequencing from RNaseR-treated RNA samples (Additional file 1: Fig. S5B). Of note, expression levels of these two circRNAs (*circZNF292-nov* and *circZNF292*) and their cognate linear RNA (linear *ZNF292*) were comparable across different tissues and cell lines in the CIRCpedia database [18], which

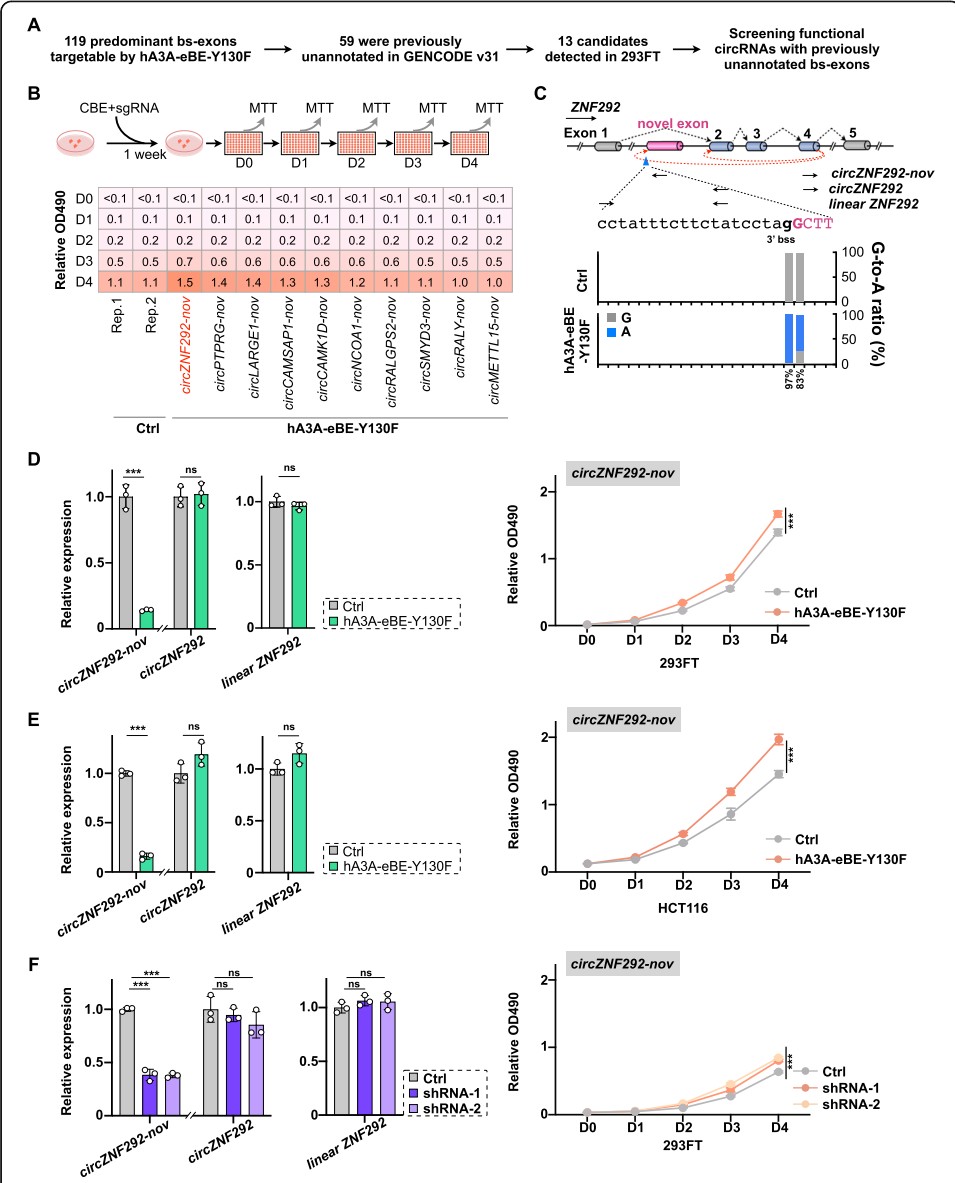

**Fig. 5** Application of BEs for functional circRNA screening. **A** Computational screening of circRNAs with predominantly back-spliced exons could be targeted by hA3A-eBE-Y130F at back-splice sites. Thirteen circRNAs identified in at least two of three (ribo−, polyA−, or RNaseR-treated) RNA-seq datasets from 293FT cells were used for functional circRNA screening. **B** Cell viability revealed by MTT assay. Top: Schematic for cell proliferation assay to detect the effect of circRNAs on the cell growth. See "Methods" for details. Bottom: $OD_{490}$ values measured at day 0, 1, 2, 3, and 4 are listed. **C** Base changes at 3′ bss of the novel exon in the *ZNF292* gene locus by hA3A-eBE-Y130F repressed the back-splice for *circZNF292-nov*. Top, Schematic of partial *ZNF292* gene organization. Back-spliced novel exon was highlighted by pink bar. Context sequences of targeted 3′ (b)ss were shown by a, t, c, and g for intron or by A, T, C, and G for exon; Bottom, G-to-A base change ratio at targeted 3′ (b)ss of predominantly back-spliced novel exon in the *ZNF292* gene locus. **D** Repression of *circZNF292-nov* by BE promotes 293FT cell proliferation, as revealed by MTT cell proliferation assays. Left, 3′ bss mutation decreases expression of *cirZNF292-nov*, but not expression of *cirZNF292* or linear *ZNF292* RNA(s). Right, cell proliferation ability revealed by MTT assays. **E** Repression of *circZNF292-nov* by BE promotes HCT116 cell proliferation. Refer to **D** for details. **F** *circZNF292-nov* knockdown by shRNAs also promotes 293FT cell proliferation, as revealed by MTT cell proliferation assays. **D–F** Error bar represents SD from three independent replicates. ns, not significant; ∗∗∗, $P < 0.001$; Student's *t* test

are enriched in different brain samples when evaluated by CIRCexplorer3-CLEAR pipeline (Additional file 1: Fig. S5C) [21]. As expected, disruption of 3′ bss at the previously unannotated exon only suppressed the expression of *circZNF292-nov*, but not *circZNF292* and *linear ZNF292* (left, Fig. 5D), further suggesting the specificity of BE and the observed cell proliferation effect by *circZNF292-nov* (right, Fig. 5D).

To examine whether the suppression of *circZNF292-nov* on cell growth is cell-type-dependent, we disrupted the 3′ bss of *circZNF292-nov* in HCT116 cells with the same strategy by hA3A-eBE-Y130F and identified that the *circZNF292-nov* depletion resulted in increased cell growth of HCT116 cells as well (Fig. 5E). Finally, treatment of 293FT cells with two shRNAs specifically targeting *circZNF292-nov* also led to downregulation of *circZNF292-nov*, which correspondingly increased 293FT cell growth, further confirming the suppressive effect of *circZNF292-nov* on cell growth (Fig. 5F). As a negative control, depletion of *cirRALY-nov* showed little effect on 293FT cell growth (Additional file 1: Fig. S5D). These results thus suggested the specificity and reliability of BEs to study the function of circRNA with predominantly back-spliced exons.

## Discussion

A significantly large number of circRNAs have been recently identified across different cell lines/tissues and across species. However, understanding their functions has just begun. Studies of biological significance of individual circRNAs have been impeded, largely due to the unavailability of effective tools that can discriminate circRNAs from cognate linear mRNAs [5]. LOF, together with gain-of-function (GOF), is commonly applied to interrogate genes′ biological significances. By introducing out-of-frame mutations with classical Cre-LoxP or modern CRISPR/Cas9-mediated genome-editing systems, LOF of linear protein-coding genes can be achieved at the protein level. However, this strategy does not work well for circRNA study due to at least two reasons. On the one hand, the same exons for many circRNAs also appear in the cognate linear RNAs; correspondingly, sequence changes at circRNAs can cause out-of-frame mutations in linear RNAs, resulting in unwanted LOF of linear RNAs. On the other hand, most circRNAs do not likely associate with polysomes for encoding functional proteins [52], and thus, it is impractical to generate un- or mis-translatable products for LOF of most circRNAs. So far, only a few cases were reported for circRNA KO by removing the entire circle-forming exon [32] or indirectly disrupting the pairing of ICSs to reduce the amount of circRNA back-splice [10]. However, these methods suffer from the depletion of large fragments in the genome that leads to a disruption of the same exons in linear RNAs or being inadequate in removing all potential ICSs at most circRNA-producing loci. It has remained a challenge to specifically and precisely target circular, but not linear, RNAs at the genomic level for reliable LOF studies [5, 53].

Here, we presented an alternative way for circRNA LOF studies by editing sequences at back-splice sites with BEs. Compared to aforementioned cases for mouse *Cdr1as/ciRS-7* (Additional file 1: Fig. S6A) [32] or human *circGCN1L1* KO (Additional file 1: Fig. S6B) [10] by CRISPR/Cas9 systems, BEs precisely introduce a few base changes at back-splice sites to obtain successful KO effect (Additional file 1: Fig. S6C), without the requirement of deleting large genomic sequences. In addition, BE-mediated nucleotide changes do not generate DNA double-strand breaks (DSBs) in genome as the CRISPR/Cas9-mediated deletion [37–40]. Moreover, by Sanger sequencing, almost no mutation

could be examined at multiple sgRNA-dependent off-target sites in *CDR1as/ciRS-7* KO monoclones (Additional file 1: Fig. S7). In this scenario, fewer side-effects were expected by using BEs to deplete circRNA biogenesis than other genome deletion methods. Of note, with the recently reported transformer BE system, the depletion of circRNA could be further achieved without introducing both sgRNA-dependent and sgRNA-independent off-target mutations [54]. Finally, a small-scale screening with BEs also led to the discovery of functional circRNAs, such as *circZNF292-nov*, to be involved in cell proliferation, while the detailed mechanism how *circZNF292-nov* suppresses cell growth is awaiting to be further explored.

A major limitation for BE-mediated circRNA KO is the concurrent influence on linear RNA splice and/or expression, while it is indeed a common disadvantage for circRNA KO by all other current methods as most highly expressed circularized exons were embedded in the middle regions of genes [16] and also involved in canonical splice for linear RNAs. Thus, base changes at exon/intron (or intron/exon) junctions of these circularized exons could theoretically affect both circular and their cognate linear RNA expression (Fig. 2, Additional file 1: S1 and S2). Similarly, it is also possible that genomic splice site mutagenesis by BEs for splicing alternation [42, 43] could also unintentionally lead to back-splicing changes. Nevertheless, to minimize this inevitable side-effect, we suggest to apply BEs to target exon(s) that are predominantly back-spliced in circRNAs (Figs. 3 and 4). In this study, we have provided lines of proof-of-principle evidence to specifically deplete circRNAs with predominantly circularized exons, *CDR1as/ciRS-7* (Fig. 3), *circRALY-nov*, *circCAMK1D-nov* (Fig. 4), and *circZNF292-nov* (Fig. 5), in human 293FT and HCT116 cell lines. Other than KO, reducing circRNA expression was also reported at the RNA level by short hairpin RNAs (shRNAs), small interfering RNAs [26, 27], or RNA-targeting type VI CRISPR effector RfxCas13 systems [55]. In all cases, shRNAs, siRNAs, or gRNAs were designed to target sequences of circRNA-featured BSJ sites for targeted circRNA repression. Comparative analyses suggested that circRNA knockdown by RfxCas13 showed much less off-target on cognate mRNAs than those by shRNAs/siRNAs [55]. Although nearly all BSJs are targetable by RfxCas13-gRNA, the execution at the RNA level of circRNA LOF depends on the continuous expression of the RfxCas13 system in cells. Differently, the BE-mediated circRNA KO is achieved permanently at the genomic DNA level, which can be used for the studies of circRNA biogenesis and function in vivo. Of note, the application of BE for circRNA KO complements the reported RfxCas13/shRNA/RNAi for circRNA KD, which together will impel the study of circRNAs.

Another obstacle of using BEs for circRNA KO is due to the PAM constraints. For example, only one third of high-confidence back-spliced exons in 293FT cells could be targeted by examined BEs, including hA3A-eBE-Y130F and ABEmax (Fig. 1). This limitation could be partially solved by using additional BEs with engineered Cas proteins to extend BE-targetable back-splice sites [40], such as replacing nCas9-NGG with nCas9-NG [56] or a near-PAMless SpCas9 variant nSpRY [57]. For example, only one or no sgRNA could be designed with hA3A-eBE3-Y130F requiring NGG PAM to target 5′ (b)ss of exon 5 in the *SPECC1* gene locus or 5′ (b)ss of exon 7 in the *ARCN1* gene locus (Fig. 6A). Instead, two or one and seven or six sgRNAs could be theoretically designed by a further engineered hA3A-eBE3-Y130F with nCas9-NG (Fig. 6B) or with nSpRY that requires NRN/NYN PAM (Fig. 6C), at corresponding sites. Importantly,

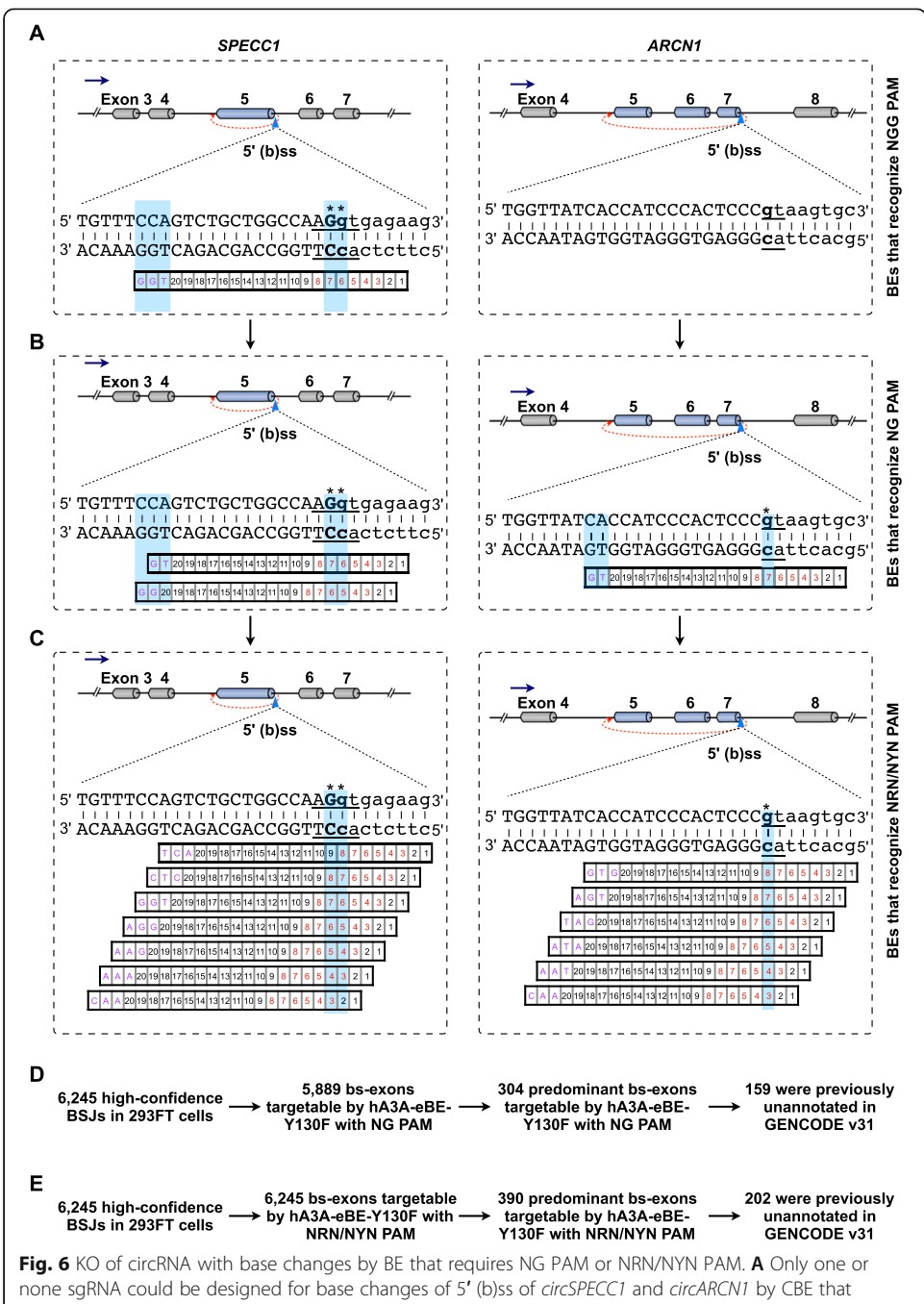

**Fig. 6** KO of circRNA with base changes by BE that requires NG PAM or NRN/NYN PAM. **A** Only one or none sgRNA could be designed for base changes of 5′ (b)ss of *circSPECC1* and *circARCN1* by CBE that requires NGG PAM. **B** Two or one sgRNA could be designed for base changes of 5′ (b)ss of *circSPECC1* and *circARCN1* by CBE that requires NG PAM. **C** Seven or six sgRNAs could be designed for base changes of 5′ (b)ss of *circSPECC1* and *circARCN1* by nSpRY-conjugated BE that requires NRN/NYN PAM. **D** In silico screening of circRNAs with predominantly back-spliced novel exons, targeted by hA3A-eBE-Y130F with NG PAM at back-splice sites. **E** In silico screening of circRNAs with predominantly back-spliced novel exons, targeted by nSpRY-conjugated BE that requires NRN/NYN PAM at back-splice sites

most (5889 or 6245 out of 6245) high-confidence BSJs identified in 293FT cells could be targetable by the engineered hA3A-eBE3-Y130F with nCas9-NG or nSpRY in silico, and thus more predominantly back-spliced exons could be selected for further BE-

mediated KO analysis by BEs allowing NG PAM or all PAM than those requiring NGG PAM (Fig. 6D and E).

## Conclusion

Collectively, by editing overlapped (back-)splice sites for both linear and circular RNA production (Fig. 2) or non-overlapped or back-splice sites predominantly for circRNA biogenesis (Figs. 3 and 4), the current study confirmed the requirement of canonical splice signals for circRNA biogenesis and further showed the applicability of BE-mediated circRNA KO for functional screening (Fig. 5). In the future, this developed BE-mediated KO strategy could be widely used for the circRNA study. For example, by introducing an early stop codon, such as mutating CAG to TAG, within a predicted open read frame of given circRNAs, this BE-mediated method can validate whether these circRNAs are translatable. In so, its corresponding protein/peptide could be depleted after this BE-mediated mutation. In addition, by mutating splice sites, which are uniquely for linear RNA biogenesis but not for back-splice, this BE-mediated method could be also used to examine the potential crosstalk and competition between linear and circular RNA biogenesis.

## Methods

### Cell culture

Human cell lines HCT116 cells were purchased from the American Type Culture Collection (ATCC; http://www.atcc.org), 293FT cells were purchased from Thermo Fisher and were originally authenticated using STR profiling. 293FT cells (human fetus origin) and HCT116 cells (human female origin) were maintained in DMEM supplemented with 10% fetal bovine serum (FBS) and 0.1% penicillin/streptomycin. We maintained cell lines at 37 °C in a 5% $CO_2$ cell culture incubator and tested all cell lines routinely for Mycoplasma contamination.

### Plasmid construction

To generate sgRNA expression vectors, oligonucleotides of sgRNA were annealed and ligated into BsaI-linearized pGL3-U6-sgRNA-PGK-puromycin (Addgene, 51133). To generate shRNA expression vectors, oligonucleotides of shRNA were annealed and ligated into AgeI/EcoRI-linearized pLKO.1-TRC (Addgene, 10878). Sequences of oligonucleotides used for sgRNA and shRNA expression vector construction were listed in Additional file 3: Table S2.

### Cell transfection and single cell cloning

Plasmid transfection was carried out with Lipofectamine 3000 Reagent (Invitrogen) according to the manufacturer's protocols. Briefly, 293FT cells were seeded in a 12-well plate at a density of $2 \times 10^5$ cells per well and co-transfected with 1.7 μg of a specific CBE, hA3A-eBE-Y130F [45] or ABE, ABEmax [46] expression vector, and 1.1 μg sgRNA expression vector per well. Transfected 293FT cells with the empty PGL3 and BE expression vectors were used as control (Ctrl). After 24 h of transfection, puromycin was added into the culture medium at a final concentration of 1 μg/ml to select

transfected cells for 3 days. After puromycin selection, cells were collected for further analyses.

To generate *CDR1as/ciRS-7* knockout stable cell lines, control or BE-treated 293FT mixture cells were digested and re-suspended in DMEM, and then plated on 96-well plates at a concentration of ~1 cell per well. Twenty monoclones were randomly selected and their genotypes were examined by PCR and Sanger sequencing to confirm base editing outcomes at the back-splice site of circularized *CDR1as/ciRS-7* exon. Of note, as 293FT cells contain three copies of X chromosomes, there are three alleles of *CDR1as/ciRS-7* gene loci. Expected base substitutions could theoretically happen at one, two, or all three alleles. Four *CDR1as/ciRS-7*-KO stable cell lines from individual monoclones were obtained for subsequent analysis.

### Lentivirus production and cell infection
To produce lentiviral particles, $5 \times 10^6$ 293FT cells were seed in a 10-cm dish for 24 h and then co-transfected with 10 μg shRNA vector, 7.5 μg psPAX2, and 3 μg pMD2.G vector. The supernatant containing viral particles was harvested twice at 48 and 72 h after transfection, then filtered through a Millex-GP filter unit (0.45 μm pore size, Millipore), and enriched by Lenti-Concentin Virus Precipitation Solution (ExCell Bio), finally in 1 ml PBS containing 0.1% BSA. Lentivirus infection was performed by culturing cells in medium containing lentivirus and 1 μg ml$^{-1}$ polybrene (Sigma), and 1 μg ml$^{-1}$ puromycin selection was used several days to increase the knockdown efficiency.

### RNA isolation, RT-PCR, RT-qPCR, and RNase R treatment
Total RNAs from cultured cells were extracted with Trizol (Life technologies) according to the manufacturer's protocol, and then treated with DNase I (Ambion, DNA-free kit) to remove genomic DNA contamination. DNase I-treated total RNAs were reverse transcribed with SuperScript III (Invitrogen) for cDNAs. Expression of each examined gene was determined by PCR/qPCR amplification of cDNAs with corresponding primers listed in Additional file 3: Table S2. Convergent primers and divergent primes are used to evaluate linear or circular RNA splice/expression, respectively. Expression of *β-actin* mRNA was used as an internal control. RNase R treatment was performed as previously described [16] for circRNA enrichment.

### Library preparation and deep sequencing
Polyadenylated and non-polyadenylated RNA separation, and RNaseR treatment were carried out as previously described [58, 59]. Briefly, total RNAs were incubated with oligo(dT) magnetic beads to isolate either poly(A)+ RNAs, which were bound to beads, or non-poly(A)+ RNAs, which were present in the flowthrough after incubation. Oligo(dT) magnetic bead selection was performed three times individually to ensure pure poly(A)+ or non-poly(A)+ RNA populations. The non-poly(A)+ RNA population was further processed with the RiboMinus kit (Human/Mouse Module, Invitrogen, Carlsbad, CA, USA) to deplete most of the abundant ribosomal RNAs to obtain poly(A)–/rRNA– RNAs (poly(A)– RNAs for simplicity). An aliquot of poly(A)– RNAs was incubated with 40 U of RNase R (Epicenter) for 3 h at 37 °C and then were subjected to purification with Trizol to obtain RNaseR-treated RNAs. All three groups of poly(A)+,

poly(A)– and RNaseR-treated RNAs were individually subject to RNA-seq library preparation by using Illumina TruSeq RNA Sample Prep Kit V2 and to then deep sequencing with Illumina HiSeq 2000 at Shanghai Institute of Nutrition and Health, CAS for Computational Biology Omics Core, Shanghai, China.

### Northern blotting (NB)

NB was performed according to the manufacturer's protocol (DIG Northern Starter Kit, Roche). In brief, 5 μg total RNAs were denatured at 95 °C for 5 min and resolved on 1.5% native agarose gel for electrophoresis, transferred to nylon membrane (Roche), and UV-crosslinked. Membrane was then hybridized with specific Dig-labeled riboRNA probes that were made using RiboMAX Large-Scale RNA Production Systems (Promega). Primers for NB probe is listed in Additional file 3: Table S2.

### Cell proliferation assay

To detect the effect of circRNA on cell growth, cell proliferation assay was performed by using the CellTiter 96® AQueous One Solution Cell Proliferation Assay (Promega) according to the manufacturer's protocol. Briefly, 7 days after hA3A-eBE-Y130F treatment, cells were trypsinized and calculated by Countess II FL Automated Cell Counter (Thermo Fisher), and seeded to a 96-well plate at a density of 3000 cells/well. At about 6 h after seeded, absorbance value ($OD_{490}$) of cell density was examined by Epoch 2 Microplate Spectrophotometer to obtain the cell proliferation value at day 0 after subtracting background absorbance. Additional cell proliferation values were examined at day 1, day 2, day 3, and day 4, respectively, and used for comparison after removing batch effects between seeded wells with different treatment cells.

### Consensus sequence analysis of splice and back-splice sites

Known human (hg38) gene annotations (human gencode.v31.annotation.gtf and refFlat.txt updated at 2017/08/23) were downloaded from GENCODE and UCSC databases. Genomic coordinates of 5′ splice site (ss) and 3′ ss of all annotated exons in these GENCODE and UCSC databases were retrieved. Human circRNA annotation, based on known human (hg38) gene annotation, was downloaded from CIRCpedia v2. Genomic coordinates of 5′ back-splice site (bss) and 3′ bss of circularized exons were retrieved from this circRNA annotation. Genomic coordinates of 5′ ss and 3′ ss of exons that are not back-spliced were also retrieved.

Ten bases around 5′ bss/5′ ss (three upstream exonic bases and seven downstream intronic bases) and ten bases around 3′ bss/3′ ss (seven upstream intronic bases and three downstream exonic bases) were fetched by bedtools (2.26.0, parameter: getfasta -s -name), and the sequence logos were drawn by R library ggseqlogo (0.1). Of note, intronic sequences were represented by a, t, c, and g, and exonic sequences were represented by A, T, C, and G.

### Calculation of base editing ratio at both on-target and off-target sites

Genomic DNAs were extracted from transfected cells with TIANamp Genomic DNA Kit (TIANGEN) according to the manufacturer's protocols. SgRNA-dependent off-target sites were predicted by the previously published Cas-OFFinder method [60].

Genomic DNA fragments of on-target and sgRNA-dependent off-target sites were individually amplified with primers listed in Additional file 3: Table S2, and further examined by Sanger sequencing. To calculate editing ratio at each on-target site or mutation ratio at each off-target site, heights of A, T, C, and G signals of Sanger sequencing were retrieved by Bioedit [61], and processed by the following equation: editing or mutation ratio = [$C_{height}/(C_{height} + T_{height})$ or $A_{height}/(A_{height} + G_{height})$]. Successful C-to-T (G-to-A) editing by CBEs or A-to-G (T-to-C) editing by ABEs was observed at targeted (b)ss. Of note, ~ 33%, ~ 67%, and ~ 100% base editing ratios indicated base substitutions at one, two, or all three alleles in 293FT cells.

## Profiling novel back-spliced exons that are predominantly processed in circRNAs

RNA-seq datasets from published ribo− [55], poly(A)−, and RNaseR-treated RNAs in 293FT cells were used for circRNA profiling. Briefly, RNA-seq fragments were mapped by HISAT2 (2.0.5; parameters: hisat2 --no-softclip --scoremin L, -16,0 --mp 7,7 --rfg 0,7 --rdg 0,7 --dta -k 1 --max-seeds 20) against the GRCh38/hg38 human reference genome with known gene annotations (gencode.v31.annotation.gtf). HISAT2-unmapped fragments were then re-aligned to the same GRCh38/hg38 reference genome using TopHat-Fusion (2.0.12; parameters: tophat2 -fusion-search --keep-fasta-order --bowtie1 --nocoverage-search) to identify high-confidence BSJ sites by CIRCexplorer2 (2.3.6) [36] with additional parameters: mapped fragments ≥ 3, containing GU/AG splice site motif with 3-nt offset, length between two splice sites ≤ 30,000 nt. Predominantly back-spliced exons spanning high-confidence BSJs were identified by requiring HISAT2-mapped fragments ≤ 3 from canonical splicing of these (back-)spliced exons in 293FT poly(A)+ RNA-seq. Predominantly back-spliced novel exons spanning high-confidence BSJs were further selected without GENCODE annotation.

## Design of sgRNAs for targeted back-spliced exons

To design specific sgRNAs for targeted BSJs, flanking regions of their back-splice sites were searched to find nearby PAM motifs that could fit the targeted bases at back-splice sites into the editing windows of used BEs, such as hA3A-eBE-Y130F or ABE-max, by previously reported BEable-GPS method [44]. Of note, both NGG and NG PAM sequences were used for this prediction.

## Small-scale screening for functional circRNAs with hA3A-eBE-Y130F

A small-scale screening was performed to identify functional circRNAs with hA3A-eBE-Y130F. Among 119 predominantly back-spliced exons in 293FT cells, 59 were previously unannotated in GENCODE reference (human gencode.v31.annotation.gtf). In addition, thirteen of these previously unannotated, predominantly back-spliced exons were successfully detected in at least two of three (ribo−, polyA−, or RNaseR-treated RNA-seq) datasets from 293FT cells and were then selected for further function screening. Corresponding sgRNAs were designed to target the novel back-splice sites of these circRNAs, and individually co-transfected to 293FT cells together with the vector for hA3A-eBE-Y130F. After 7 days, these BE-edited cells were applied for cell proliferation analysis.

## Supplementary Information

---

**Additional file 1:** Supplementary figures. Fig. S1 Base changes at back-splice sites by BE4max or eBE-S3 (Related to Fig. 2). Fig. S2 Base changes at back-splice sites by ABEmax (Related to Fig. 2). Fig. S3 Validation of *circRALY-nov* and *circCAMK1D-nov* (Related to Fig. 4). Fig. S4 Thirteen circRNA KO by base changes (Related to Fig. 5). Fig. S5 Functional analysis of *circZNF292-nov* and *circRALY-nov* (Related to Fig. 5). Fig. S6 Overview of current strategies for circRNA knockout (Related to Fig. 6). Fig. S7 Examination of editing ratios of on-target sites and mutation ratios at selected gRNA-dependent off-target sites in *CDR1as/ciRS-7* KO and negative control monoclones

**Additional file 2:.** Table S1. List of high-confidence circRNAs in 293FT Cells. High-confidence circRNAs was determined from ribo−, poly(A)− and RNaseR-treated RNA-seq in 293FT cells, shown by circRNA location, strand, gene symbol, transcript ID, included exons, FPB in ribo−, p(A)− and RNaseR-treated samples, whether can be targeted by hA3A-eBE-Y130F or ABEmax, whether have predominantly bs-exons, and whether have novel bs-exons

**Additional file 3:.** Table S2. List of oligonucleotides and primer sequences used in this study. (A) Sequences of oligonucleotides used in sgRNA constructs. (B) On-target primer sequences used in genomic DNA amplification. (C) The gRNA-dependent off-target primer sequences used in genomic DNA amplification. (D) Primer sequences used in RT-qPCR and RT-PCR analysis. (E) Primer sequences used in NB. (F) Sequences of oligonucleotides used in shRNA constructs

**Additional file 4:.** Review history

---

### Acknowledgements
We thank Yang and Chen laboratories for discussion.

### Review history
The review history is available as Additional file 4.

### Peer review information

### Authors' contributions
L.Y. supervised and conceived the project. L.Y., L.-L.C., X.G., and X.-K.M. designed experiments. X.G., X.L., C.-X.L., J.Z., J.W., and Y.W. performed experiments, supervised by L.Y. and L.-L.C; X.-K.M. and G.-W.L. performed computational analyses, supervised by L.Y.; J.C. participated in data interpretation. L.-L.C. and L.Y. wrote the paper with input from X.G. and X.-K.M. All author(s) read and approved the final manuscript.

### Funding
This work was supported by grants from Ministry of Science and Technology of China, China (2019YFA0802804, 2021YFA1300503) and the National Natural Science Foundation of China, China (31925011, 31730111, 91940306) to L.Y, the National Natural Science Foundation of China, China (31725009) and the Howard Hughes Medical Institute International Program, USA (55008728) to L.L.C., China National Postdoctoral Program for Innovative Talents (BX2021337), Shanghai Post-doctoral Excellence Program, China (2020506) and Special Research Assistant of the Chinese Academy of Sciences, China to X.K.M, National Natural Science Foundation of China, China (32101038) to X.L.

### Availability of data and materials
The accession number for poly(A)+, poly(A)−, and RNaseR-treated RNA-seq datasets from 293FT cells is GEO: GSE172193 [62] and NODE: OEP002843 [63]. The rRNA-depleted (ribo−) RNA-seq datasets of 293FT cell line were downloaded from NCBI GEO (GSE149691).

## Declarations

### Ethics approval and consent to participate
Not applicable.

### Consent for publication
Not applicable.

### Competing interests
L.L.C. and X.L. have filed a patent application relating to the published work through CAS Center for Excellence in Molecular Cell Science, Shanghai Institute of Biochemistry and Cell Biology, Chinese Academy of Sciences. However, the patent does not restrict the educational, research, and not-for-profit purposes. The other authors declare that they have no competing interests.

### Author details
[1]State Key Laboratory of Molecular Biology, Shanghai Key Laboratory of Molecular Andrology, CAS Center for Excellence in Molecular Cell Science, Shanghai Institute of Biochemistry and Cell Biology , University of Chinese Academy of Sciences, Chinese Academy of Sciences, 320 Yueyang Road, Shanghai 200031, China. [2]School of Life Science and Technology, ShanghaiTech University, 393 Middle Huaxia Road, Shanghai 201210, China. [3]CAS Key

Laboratory of Computational Biology, Shanghai Institute of Nutrition and Health , University of Chinese Academy of Sciences, Chinese Academy of Sciences, 320 Yueyang Road, Shanghai 200031, China. ⁴Hangzhou Institute for Advanced Study, University of Chinese Academy of Sciences, Hangzhou 330106, China. ⁵CAS Center for Excellence in Molecular Cell Science, Shanghai Institute of Biochemistry and Cell Biology, Chinese Academy of Sciences, Shanghai 200031, China.

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

## 
