## [**Additional file 4:.** Review history · Genome Biology]

Review History

First round of review

Reviewer 1

Were you able to assess all statistics in the manuscript, including the appropriateness of statistical tests used? Yes.

Were you able to directly test the methods? Yes.

Comments to author:

Gao et al., Li Yang and colleagues

"Knockout of circRNAs by base editing back-splice sites of circularized exons"

Gao et al. from Li Yang's and Lingling Chen's group at the CAS Shanghai Institute of Biochemistry and Cell Biology describe here a systematic study of a knockout strategy by base-editing splice sites of circularized exons, with the goal to specifically knockout circRNA generation without affecting corresponding linear RNA products. In contrast to the very few, CRISPR-Cas-based knockout studies done recently, this approach changes only single nucleotide positions at splice sites, that are exclusively used in back-splicing events. As a result, in cases, where there is exclusive back-splicing or where certain back-splice-specific exons occur, this will provide a valuable strategy.

The general concept (see section in Results on "Design"), the validity, potential, and limitations of this approach (see Discussion) are clearly described, controlled, and carefully presented in this manuscript. Different editing systems are used and compared with each other, partly also in different cell lines. Finally, a small-scale screen was set up, testing 13 circRNAs with exons specifically used in back-splicing, which are amenable to this approach. Although only useful for a relatively small subset of circRNAs, it represents a valuable, state-of-the-art approach in addition to existing tools to investigate circRNAs at the functional level, which is of importance and sufficient general interest in the circRNA field, and which should be published.

Reviewer 2

Were you able to assess all statistics in the manuscript, including the appropriateness of statistical tests used? No.

Were you able to directly test the methods? Yes.

Comments to author:

The manuscript presented by Gao and colleagues describes a method for circRNA-specific knockout by base editing. In the study, they demonstrated BEs repressed both circular and linear RNAs expression by targeting splice sites involved in both back-splicing and canonical splicing. In addition, they showed BEs can specifically knock out circRNA expression by targeting back-splice sites predominantly for circRNA biogenesis. They further applied BEs for functional circRNA screening. Overall, the manuscript is well designed and exhibited comprehensive work, and provide significant data supporting their conclusions. However, a few improvements should

be considered before it can be finalized for publication.

1. Although fewer side-effects were expected by using BEs to deplete circRNA biogenesis than other genome deletion methods, as discussed in the manuscript, the authors should predict and detect several off-target sites for tested targets, and evaluate the sgRNA-dependent off-target efficiency of BEs for circRNAs KO.

2. In the fourth paragraph of the "Discussion and conclusion" section, the authors proposed to replace nCas9-NGG with nCas9-NG to target more sites for circRNA KO. As reported recently, a near-PAMless SpCas9 variant SpRY was developed, which can target almost all PAMs (Walton RT et al., 2020, Science). I think it will be better to add nSpRY in the discussion.

We like to thank reviewers for your general support and positive comments on this manuscript. We also appreciate their insightful suggestions that have guided us to further improve this manuscript. Please find our point-by-point responses below.

Reviewers' comments:

Reviewer #1:

Gao et al. from Li Yang's and Lingling Chen's group at the CAS Shanghai Institute of Biochemistry and Cell Biology describe here a systematic study of a knockout strategy by base-editing splice sites of circularized exons, with the goal to specifically knockout circRNA generation without affecting corresponding linear RNA products. In contrast to the very few, CRISPR-Cas-based knockout studies done recently, this approach changes only single nucleotide positions at splice sites, that are exclusively used in back-splicing events. As a result, in cases, where there is exclusive back-splicing or where certain back-splice-specific exons occur, this will provide a valuable strategy.

The general concept (see section in Results on "Design"), the validity, potential, and limitations of this approach (see Discussion) are clearly described, controlled, and carefully presented in this manuscript. Different editing systems are used and compared with each other, partly also in different cell lines. Finally, a small-scale screen was set up, testing 13 circRNAs with exons specifically used in back-splicing, which are amenable to this approach. Although only useful for a relatively small subset of circRNAs, it represents a valuable, state-of-the-art approach in addition to existing tools to investigate circRNAs at the functional level, which is of importance and sufficient general interest in the circRNA field, and which should be published.

We thank this reviewer for his/her positive comments and general support.

Reviewer #2: The manuscript presented by Gao and colleagues describes a method for circRNA-specific knockout by base editing. In the study, they demonstrated BEs repressed both circular and linear RNAs expression by targeting splice sites involved in both back-splicing and canonical splicing. In addition, they showed BEs can specifically knock out circRNA expression by targeting back-splice sites predominantly for circRNA biogenesis. They further applied BEs for functional circRNA screening. Overall, the manuscript is well designed and exhibited comprehensive work, and provide significant data supporting their conclusions. However, a few improvements should be considered before it can be finalized for publication.

We thank this reviewer for his/her positive comment and general support, and have now revised this manuscript according to the reviewer's suggestion.

1. Although fewer side-effects were expected by using BEs to deplete circRNA biogenesis than other genome deletion methods, as discussed in the manuscript, the authors should predict and detect several off-target sites for tested targets, and evaluate the sgRNA-dependent off-target efficiency of BEs for circRNAs KO.

Thanks for this constructive suggestion. In this revised manuscript, we have predicted the sgRNA-dependent off-target sites according to the previously-published Cas-OFFinder (Bae S et al., 2014, *Bioinformatics*, PMID: 24463181) and further examined mutation ratios of multiple predicted sgRNA-dependent off-target sites in *CDR1as/ciRS-7* KO monoclones (new Fig. S7). Different to high editing ratios at the on-target sites, no mutation ratio could be examined

at these sgRNA-dependent off-target sites, further suggesting the high specificity of BE-mediated circRNA method.

2. In the fourth paragraph of the "Discussion and conclusion" section, the authors proposed to replace nCas9-NGG with nCas9-NG to target more sites for circRNA KO. As reported recently, a near-PAMless SpCas9 variant SpRY was developed, which can target almost all PAMs (Walton RT et al., 2020, Science). I think it will be better to add nSpRY in the discussion.

Thanks. In this revised manuscript, we have added nSpRY in the discussion, with the addition analysis of targeted editing sites by nSpRY-conjugated BE that requires NRN/NYN PAM (new Fig. 6C and 6E) and the advantage of using this nSpRY-BE for circRNA KO.